# Crack-Bridging Property Evaluation of Synthetic Polymerized Rubber Gel (SPRG) through Yield Stress Parameter Identification

**DOI:** 10.3390/ma14247599

**Published:** 2021-12-10

**Authors:** Jong-Yong Lee, Hyun-Jae Seo, Kyu-Hwan Oh, Jiang Bo, Sang-Keun Oh

**Affiliations:** 1Doctorial Course of Department of Architecture, Seoul National University of Science & Technology, 232 Gongneung-ro, Nowon-gu, Seoul 01811, Korea; jylee@re-new.co.kr; 2Doctorial Course of Convergence Institute of Biomedical Engineering and Biomaterials, Seoul National University of Science & Technology, 232 Gongneung-ro, Nowon-gu, Seoul 01811, Korea; hyjay@nate.com; 3Institute of Construction Technology, Seoul National University of Science & Technology, 232 Gongneung-ro, Nowon-gu, Seoul 01811, Korea; kyuhwan.oh@seoultech.ac.kr; 4School of Civil Engineering Architecture and Environment, Hubei University of Technology, No. 28, Nanli Road, Hongshan District, Wuchang, Wuhan 430068, China; jiangbo15@126.com; 5School of Architecture, Seoul National University of Science & Technology, 232 Gongneung-ro, Nowon-gu, Seoul 01811, Korea

**Keywords:** stress-strain analysis, grout injection, leakage crack, waterproofing, synthetic polymerized rubber gel

## Abstract

Yield stress parameter derivation was conducted by stress-strain curve analysis on four types of grout injection leakage repair materials (GILRM); acrylic, epoxy, urethane and SPRG grouts. Comparative stress-strain curve analysis results showed that while the yield stress point was clearly distinguishable, the strain ratio of SPRG reached up to 664% (13 mm) before material cohesive failure. A secondary experimental result comprised of three different common component ratios of SPRG was conducted to derive and propose an averaged yield stress curve graph, and the results of the yield stress point (180% strain ratio) were set as the basis for repeated stress-strain curve analysis of SPRGs of up to 15 mm displacement conditions. Results showed that SPRG yield stress point remained constant despite repeated cohesive failure, and the modulus of toughness was calculated to be on average 53.1, 180.7, and 271.4 N/mm^2^, respectively, for the SPRG types. The experimental results of this study demonstrated that it is possible to determine the property limits of conventional GILRM (acrylic, epoxy and urethane grout injection materials) based on yield stress. The study concludes with a proposal on potential application of GILRM toughness by finite element analysis method whereby strain of the material can be derived by hydrostatic pressure. Comparative analysis showed that the toughness of SPRG materials tested in this study are all able to withstand hydrostatic pressure range common to underground structures (0.2 N/mm^2^). It is expected that the evaluation method and model proposed in this study will be beneficial in assessing other GILRM materials based on their toughness values.

## 1. Introduction

Synthetic polymerized rubber gel (SPRG) is a grout injection leakage repair materials (GILRM) that has non-curable property intended to provide crack-bridging performance, otherwise known as ‘self-healing’ property [1]. GILRM refers to a method commonly known as injecting a liquid type or low viscosity waterproofing material into cracks on a concrete substrate whereby the material will cure or react with different reagents to form a new waterproofing layer to prevent further hydrostatic penetration [1]. In this regard, GILRM with long-term durability in concrete structures is crucial for sustainability and durability. A commonly documented problem with GILRM is that after the repair installation, there are many cases in which GILRM degrade due to the behavioral movement of the structure (crack movement), continuous water pressure, temperature change, and chemical components contained in water, resulting in material property changes in most types of GILRM [2]. One of the reasons why this problem persists is because of a lack of consistent evaluation method. Existing material property test methods in ASTM, KS, BS EN do not differentiate non-Newtonian materials with self-healing properties such as SPRG from other waterproofing materials in the market [2]. This results in a selection of GILRM during construction that is not suitable for the construction environment. GILRMs of different properties in the market are not designed to respond to all forms of deterioration environment conditions affecting leakage crack sufficient repair effect cannot be obtained [3].

In order to reduce trial and error in this field, ISO TC 71 of the International Organization for Standardization (ISO), ISO TR 16475, ISO TS 16774 was enacted to refer to leakage crack maintenance [4]. ISO TR 16475 currently proposes four types of GILRM; epoxy resin (ER), urethane resin (UR), acrylic resin (AR), and SPRG as the most effect repair materials [5]. Mohamed A. Safan investigates the effectiveness of polyurethane resin grout injection material based on its ability to prevent hydrostatic pressure [6]. Kim Soo-yeon based on ISO TS 16774 states “As a result of evaluating 4 types of GILRM for 6 types of deterioration environmental conditions, it was revealed that “crack movement behavior” had the greatest influence on the waterproofing performance of GILRM [7]. Jiangbo demonstrated the same based on the experimentation of 4 types of GILRMS [8]. As a result of evaluating the comprehensive waterproofing performance including cracking behavior for 6 types of deterioration environment, it is suggested that “synthetic rubber” has relatively superior waterproofing performance compared to other GILRMs [9]. Through these studies, it was found that the GILRM should primarily have material properties that do not break in the cracking behavior, and this study was designed to derive an evaluation methodology and criteria for assessing a GILRM’s crack-bridging performance.

In this study, an experimental regime was designed to compare and contrast the viscoelastic property and the adhesive/cohesive strength of four GILRMs (epoxy resin (ER), urethane resin (UR), acrylic resin (AR), and SPRG) through a comparative stress-strain analysis. This preliminary stage derives a potential new evaluation criteria on ‘repeatable’ yield stress property that is found unique to SPRG. Based on these criteria, a secondary repeated stress-strain analysis through a standardized adhesion strength test in compliance to KS standard that tests the yield stress limit of the SPRG is conducted. The results show SPRG that exceed the yield stress limit (cohesive failure) and subsequently put back together will produce another yield stress point. Under tensile force application condition, cohesive failure occurs before adhesive failure. This indicates that when under the effect of water pressure, the stress will primarily be generated on the body of the SPRG rather than at the cross section of the adhesion interface on the concrete surface. Based on this factor, an evaluation criteria system based on the toughness of SPRG derived from stress-strain analysis can be formulated. This is demonstrated at the end of the study by a simple finite element method analysis of an SPRG sample placed under a 0.2 N/mm^2^ hydrostatic pressure, and the result of this experiment is intended to provide a quantifiable report and data on viscoelastic GILRM materials such has SPRG on crack movement resistance property.

## 2. Objectives

### 2.1. Non-Newtonian Fluids and Viscoelastic Properties of GILRM Materials

GILRM are commonly manufactured or mixed on-site by processing solid polymers and liquid polymers according to the respective purpose and conditions. GILRMs utilize specific types of polymers, where the molecular size and structure of a high molecular liquid are complex. In contrast with a general low molecular liquid, most GILRM materials exhibit nonlinear behavior as the amount of deformation or stress increases [10]. While some variations may differ depending on the manufacturers, in most cases AR, ER, and UR materials are non-Newtonian fluids of different characteristics and elastic properties [11]. However, SPRG is considered as a viscoelastic material of Maxwell model with polymers that are in entanglement state between molecules [12]. The entangled structure of such a polymer material is formed by chemical cross-linking and is known to exhibit viscoelastic behavior as the creation and destruction of the structure are repeated as a temporary structure by physical bonding rather than a permanent form [13].

Elastomeric materials do not store energy before an irreversible deformation before external force is applied and when deformation occurs by external force [14]. For solid polymer materials, they are restored to original state even when an external force is applied, but cohesive failure will occur when the material exceeds the specified yield stress limit [15]. Such material failure is most prone to occur for GILRMs as leakage cracks are affected by repeated micro-behavioral movements caused by temperature changes in structures, earthquakes, passing vehicle loads, and vibrations. In order to respond to the effects of repetitive behavior of concrete cracks, it is necessary for GILRMs to have some degree of elastic property, while being able to maintain high surface adhesion.

Under normal testing environment, every GILRM tested under standard ASTM, KS or BS EN methods seem to comply with the above required conditions. However, there are key inconsistencies that must be addressed; (1) GILRM material properties are not so easily classified and quantifiably compared especially as they are most commonly non-Newtonian and viscoelastic materials and (2) installation, workability conditions, workmanship quality and environmental degradation factors are far too complex and varied to include in every experimental regime [16]. These factors are what determines a failed or successful installation of GILRM in leakage cracks, but as testing conditions and field application conditions are different, it is difficult to maintain a reliable report on these issues [17]. Figure 1 below provides a simplified summary of common tendencies that relate to problems caused by either poor workmanship and status after exposure to environmental degradation, among which a key parameter is crack movement (it must be noted that these tendencies are not mutually exclusive to one another and is based on a study report of sites in Korea. Cases may differ for different countries). For plastic GILRM materials that cure (harden) such as urethane (polyurethane foam) or epoxy resin, cohesive failures can occur, but more common defects will come in the form of adhesion failure. This is either due to installing on wet concrete surface or improper surface treatment prior to application. GILRM type acrylic resin are more gelatin rather than plastic in terms of their physical characteristics, and therefore most AR defects are commonly related to cohesive failure (but not exempt from adhesion failure). SPRG also show trends of cohesive failure and aging, but as SPRG is a viscoelastic material, rejoining at the cohesive failure interface is possible [18].

### 2.2. Modulus of Toughness vs. Adhesion Strength against Water Pressure as a Means of Deriving an Evaluation Criteria for Viscoelastic GILRM

Viscoelastic materials such as SPRG react differently in tensile and compressive loading. As is commonly known, polymeric deformation not only a function of applied load, but also a function of loading rate [19]. As is the case, SPRG deformation depends on time and have both the properties of solid and fluid [20]. For SPRG, molecules are held together by cohesive forces and when subject to external stress, localized rearrangement of these molecules occur [21,22]. Cross-linking and chain structure also affects the overall behavioral response of the material; when SPRG material is subject to stress beyond its yield stress, it will result in only partial recovery. References show that viscoelastic behavior can be represented by various combinations of spring and dash-pot elements in series or parallel [23], and further analysis has indicated that SPRG follows the same pattern in terms of response to stress. Due to these factors, the study aims to experimentally assess the (1) stress limits by strain-stress curve analysis of SPRG to determine whether the material is capable of maintaining crack bridging property during repeated crack movement, and (2) during and after recovery of material state, whether the SPRG is maintaining durability by measurement of toughness in relation to the expected water pressure. Water pressure is defined as force per unit area, and the amount of force that influences the SPRG material performance needs to be compared in order to determine the application of toughness against hydrostatic pressure [24,25]. As hydrostatic pressure applicable to underground structures can also be calculated by measure of energy per unit volume, as long as the strain capacity can be identified for the SPRG by the derivation of yield stress, reverse calculation would allow the derivation of maximum amount of hydrostatic pressure that can be resisted by the SPRG material (e.g., this is applicable for finite element method analysis for response to hydrostatic pressure which will be presented in the coming sections). In this particular case, assuming that the water is enclosed in the underground soil acting on the repaired concrete crack with GILRM, the hydrostatic pressure can be calculated as the following:
(1)p=w·h
where,

*p* = pressure in liquid (g/mm^2^, N/mm^2^),*w* = unit weight of water (g/mm^3^),*h* = depth of water (mm).

An experimentally determined fact about water pressure is that pressure is exerted equally in all directions. When hydrostatic pressure is being applied to the SPRG due to the presence of crack in concrete that forms a leakage path, there is a net force from the pressure that is applied perpendicular to the exposed surface of the SPRG. Pressure is not defined based on a specific direction aside from by gravity [26], meaning in this case, if there is a point on a GILRM installed concrete structure that is more easily subject to fracture and adhesive/cohesive failure, then the force from the water pressure will localize at that point. This concept is illustrated in Figure 2.

## 3. Materials and Methods

For the comparative analysis, 4 different types of GILRM were used for testing. Epoxy resin (ER) has strong adhesive strength because it is formed into a hard repair material by chemical reaction between the main agent and the curing agent but does not have flexibility and viscosity. Acrylic resin (AR) or urethane resin (UR) has semi-rigid or soft properties that harden or expand by using water as a part of the curing agent. SPRG does not mix with a curing agent or water, and its constituent components do not chemically bond, so it maintains the uncured viscosity. The respective material specifications of the tested GILRMs are outlined in Table 1 below.

SPRG used in the experiment for this analysis was a synthetic rubber-based material obtained by thermally fusion of waste oil and waste rubber. SPRGs high viscoelastic material manufactured into liquid rubber by finely pulverizing the collected and processed waste rubber for recycling to make powder with a particle size of 200 to 400 μm, and then thermally fused with waste oil. In the case of Korea, a standardized composition ratio of SPRG has not been developed [27] and this leads to case of SPRG products with different rubber mixture ratios being used in the market. While this does not fundamentally change the characteristic of SPRG, higher viscosity has traditionally proven to provide higher adhesion strength but more difficult workability and vice versa. As there is a large range of these compositions, for the experimental group in this study, the contents of waste oil and waste rubber, which are the main materials that give the adhesiveness of synthetic rubber materials, were 4:1 (low viscosity, 1.8 million cp), 3:1 (medium viscosity, 3.5 million cp), and 2:1 (high viscosity), 5 million cp in accordance to 3 most commonly used SPRG products currently being used in Korea. Material composition ratio (%) and viscosity (cp) of each type are shown in Table 2 below.

## 4. Results

### 4.1. Adhesive Strength Test Results Comparison for Four Types of GILRM

The adhesive strength of the repair material is shown in Table 3, and the images of the specimens before and after testing are shown in Figure 3. The respective stress-strain diagram for each GILRM specimen types based on the measurement are shown in Figure 4. It can be seen from the results below that the adhesive strength of the four GILRM appears in the order of AR < SPRG < UR < ER, but maximum elongation rate is in the order of SPRG > AR > UR > ER.

The overall adhesive strength of UR, ER, AR and SPRG are 0.0381, 0.6537, 0.0048 and 0.0162 MPa, respectively. For each material, the yield stress point in terms of displacement to strain ratio are as follows; for UR, at the displacement point of about 2 mm (strain rate of about 94%); for ER, at the displacement point of about 1 mm (strain ratio of about 51%); for AR, at a displacement point of about 2 mm (strain rate of about 97%); and for SPRG, at the displacement point of about 13 mm (strain of about 664%)

For ER stress-strain curve graph, a nearly vertical stress-strain graph was shown (relative to other material results), and as shown in Figure 4e. Due to the characteristics of the ER material, the adhesion between foreign substances is very high, but the elongation rate is very low, so the maximum stress appears at the minimum deformation.

For the UR stress-strain curve graph, the curve started with a steep slope, showed the maximum stress at approximately 28% strain ratio, then showed a gradual decrease and subsequent material failure. As UR is a material property that reacts and hardens with water and produces network of air pockets in the layer matrix, it is predicted that the tensile force cannot be evenly transmitted due to the open-cell network (foamed interior).

For the AR, the results were similar with UR in that failure occurred at the attachment interface, and showed that AR is possible to reach a strain-ratio of up to 97% on average.

For the SPRG, the material thinned at the cross-sectional area after maximum displacement, and cohesive failure could be observed. As a result, it was found that large stress appeared only in some initial sections, and minimal stress appeared uniformly until fracture in subsequent sections. Through the adhesion tensile performance test on these four GILRM, it was confirmed how much force each GILRM material would be able to respond to under high degrees of repeated crack movement. Following this result, a secondary experimental was conducted on the SPRG material

### 4.2. Adhesive Strength Test Results Comparison for SPRG Types (by Mixture Ratio)

For the experimental conditions, the three types of SPRG specimen differentiated by their mixture ratios in Section 3, Table 2, were tested at an experimental temperature of 20 °C. Three specimens for each type (henceforth to be identified as Type A, B and C) were tested. As a result of the experiment, the average adhesive force of Type A with a waste rubber content of 6% was 0.0136 N/mm^2^, Type B with a 7.5% content was 0.0186 N/mm^2^, and Type C with a 10% content was 0.0216 N/mm^2^. The elongation at break was 695% for Type A, 811% for Type B, and 1667% for Type C. The results summarized in Table 4 below;

#### 4.2.1. Stress-Strain Curves Analysis of SPRG

A stress-strain curve was calculated based on the measurement results of adhesive strength and elongation ratio of Types A to C. The reason for a comprehensive comparison of SPRG types of different viscosities is due to the current factor related to a lack of standardization of SPRG mixture composition. As has been explained in Section 3, higher rubber composition ratio that results in higher adhesive strength has not always resulted in positive results in terms of overall installation quality of SPRG materials. As a means to incorporate the variation of different types and viscosity ranges of SPRG materials, this study will tentatively propose a yield stress curve analysis based the average of the 3 different types of SPRG. It must be noted that further investigation by a more standardized composition of SPRG will provide different yield stress curve limit in future studies.

Stress was calculated by dividing the force by their respective section length of material deformation, and strain was calculated by dividing the elongation by the initial length of the specimen. The initial length of the specimen was based on the point at which tensile force started to increase. The maximum stress, displacement, and fracture displacement were different according to the viscosity conditions, but the stress-strain graph showed a similar shape.

An individual graph was made using each measurement data, and the point where the slope changes, that is, the position where the inflection points occur on the graph for each specimen of the SPRG types was designated as using shaped points (a triangle for the first specimen, and a square and circle subsequently) throughout the slopes of the graphs. The sequential inflections were color coded (explained the in the legend below) to clearly demarcate at which point of the stress-strain curve the inflection changes are occurring. Each of the points have been highlighted in respective circles and can be seen in Figure 5. The stress and strain at each point were extracted and shown in Table 5 and the averaged yield stress results are shown in Table 6.

In the process of designating the inflection point (yield stress), horizontal and S-curve graphs were similarly generated at the initial stress occurrence time from the 1st to the 3rd inflection points (red to yellow in Figure 5). After this point, it increased close to a straight line up to the point of maximum stress, and the stress decreased after the point of maximum stress. The graph of the linear regression line was maintained, then the highest point was reached (green in Figure 5). Subsequently, the stress remained close to zero and was maintained at such rate (blue to purple) until cohesive failure of the SPRG (cohesive failure point is not displayed on Figure 5 as the strain ratio reached ranges of approximately 1000% or above as is shown in Table 3 above).

The average stress and strain at the 4th inflection point, the average proportional limit of Type A, were 0.0126 N/mm^2^ and 67%, respectively, and the average stress and strain at the 5th inflection point, the yield point, were 0.0136 N/mm^2^ and 70%, respectively. For Type B; 0.0169 N/mm^2^ and 77% and, 0.0186 N/mm^2^ and 81%. For Type C, 0.0205 N/mm^2^ and 84% and 0.0216 N/mm^2^ and 95%.

For the comprehensive results, the stress and strain at the average proportional limit were 0.0186 N/mm^2^ and 78%, respectively, and the stress and strain at the average yield point were 0.0198 N/mm^2^ and 90%, respectively.

A series of linear regression graphs were derived obtained by graphing the inflection point extraction values in Table 4 and Table 5, and are shown in Figure 6. A comprehensive trend graph based on the average linear regressions of Type A, B, and C are provided in Figure 6d.

#### 4.2.2. Yield Stress Derivation of SPRG (Tentative Results)

As a result of the experiment, Type C formed an overall high average value, and Type A and Type B showed less than average stress and strain. This is a difference in viscosity due to the difference in the content of constituents in the material mixture, and as the viscosity increases, the cohesive force interacting between the materials also increases, indicating that the adhesive stress is increased.

The difference in viscosity was confirmed that the cohesive force between the materials was directly expressed as the maximum stress. The range of strain (resistance) due to cohesive force did not exceed 180%. It was confirmed that as the viscosity of the material increases, the point at which the maximum stress develops is delayed to the second half of the strain ratio range. In the overall averaged stress strain curve graph is provided in Figure 6 below, derived by division of sections based on the inflection point patterns. The area of the graph was largely divided into a cohesive area and a minimum cohesive force maintenance area, and the details are shown in Figure 7. (Here it is noted that the yield stress point and stress-strain curve graph is a tentative representation of SPRG materials, and is subject to change in future when repeating this evaluation process again with a standardized mixture composition of SPRG).

a.Viscoelastic material property section (Viscoelastic section): this section is compliant to conventional viscoelastic material that resists the application of external energy in the entangled state in which the coil state is maintained similar to an intermolecular skein between the crosslinking points. This section is comprised of the conventional creep property of viscoelastic materials and tensile stress application (a and b of Figure 7)b.Tensile stress application section (Tensile stress start): The crosslinking point and the intermolecular entanglement state are converted to an unwinding state as they are stretched by an external force, but a region in which deformation occurs constantly by resisting external force only with cohesive force.c.Tensile stress resistance section (SPRG resistance section): The crosslinking point and the intermolecular entanglement state are converted to an unwinding state while being stretched by an external force, and the deformation occurs constantly while maintaining the minimum cohesive force by adapting to the external force. The intermolecular entanglement between bridge points is released against external forces that continue after the yield point, but the cohesive force is maintained in a state where the entanglement state is not broken. As the SPRG material enters this section during the stress strain curve, the material shows proportional elastic behavior and yield at the maximum point of cohesive force.d.Dashpot region according to Maxwell model (particle separation section): After the resistance section, the entangled force between molecules is lost due to external force, and the bond begins to completely unravel, and the phase changes and elongates. (However, the material attached to the surface to be adhered is maintained the same as the initial adhesion area)e.Elongation section: A section where the cohesive force converges to 0 while maintaining the minimum resistance due to entanglement between molecules and breaks at the point of maximum elongation.

### 4.3. Repeated Yield Stress Limit Test Result of SPRG (Tentative Results)

Based on the derived yield stress results of the 3 compositions of SPRG, a new experimental set up was devised and executed, where the displacement condition, based on the strain ratio at the yield stress point (180%) and the correlating displacement was (13 mm) was set past the limit to induce cohesive failure of the specimen by setting the displacement to 15 mm. Upon returning to original position, the tensile stress application was repeated to a maximum of 5 times. This was conducted on the 3 different compositions of the SPRG up to 3 times. Example results are shown in Figure 7 below for the different SPRG types, respectively.

As evidenced in Figure 8a, the displacement range of 15 mm was not sufficient to induce a clear cohesive failure due to the low viscosity property of Type A SPRG. In the cases of Type B and C however, the displacement range of 15 mm was enough to induce a cohesive failure of the specimen. Once the attachment pieces with the specimen remainders returned to the original position and the separated specimen made contact, the second, third and the following cycles of tensile stress produced lower yield stress point than the first cycle, indicating that the first cohesive failure affected the durability of the SPRG material. However, after subsequent failures, yield stress point remained consistent. This result is the key indication of the SPRG’s viscoelastic property that allows it to respond to repeated cycles of concrete crack movement after installation. When calculating the toughness of SPRG can be determined by integrating the stress-strain curve. It is the energy of mechanical deformation per unit volume prior to fracture. The mathematical description is:
(2)$$U_{T}=\frac{P}{A}\times \frac{∆L}{L}=\sigma \varepsilon$$
where,

*U*_T_ = energy (Jmm^−3^, N/mm^2^)*P* = Force (N)*A =* Area (mm^2^)∆L = Change in *L* (mm)*L* = Change in Length (mm)ε= strain (unitless, (%) when considered as ratio)σ= stress (MPa, N/mm^2^)

Refer to Table 7 below for results on toughness calculation results based on the repeated tensile stress testing.

### 4.4. Possible Application of Proposed Evaluation Methodology (e.g., Finite Element Method Analysis)

By referencing this principle, the energy exerted by water pressure (calculable by Equation (1) in Section 2.2) based on the depth of the concrete structure can be calculated and be employed as a criteria index for the tested SPRG; as long as the toughness (energy) of the SPRG is higher than that of the expected water pressure force, the said material can be used, and higher toughness index can be used as an index of waterproofing performance of the SPRG. This is applicable for finite element method (FEM) analysis (via Abaqus) by the method of static hydrostatic pressure simulation of a viscoelastic model of an SPRG sample. While further analysis is still required, based on the FEM analysis, the types of SPRG can be subject to a fixed hydrostatic pressure compliant to the pressure levels common to underground structure environment (in the case of Korea, 20 m underground equates to 0.2 N/mm^2^) (Refer to Figure 9 below for illustration of the modelling samples), which when applied upon the model based on the dimensions of the specimen size (40 × 40 × 2 mm^3^), the derived strain on the material can be used to calculate the corresponding stress based on the results of Figure 8. In this sample case, based on the results of a sample modelling of Type A SPRG, the maximum deformation derived at the edges (shown as red sections in the image) is converted to a unit of 1.265 mm, where in comparison to the yield stress range, the material would not have undergone cohesive failure due to the given hydrostatic pressure.

## 5. Conclusions

This study was conducted to propose a new evaluation method for GILRM based on stress-strain analysis using standard adhesion strength testing method. The following conclusions were drawn:(1)A preliminary testing consisting of adhesive strength measurement testing of four types of GILRM (UR, ER, AR and SPRG) was conducted. Adhesive strength of the four GILRM appears in the order of AR < SPRG < UR < ER. A Stress-strain diagram for each GILRM specimen types based on the measurement results, whereby it was confirmed that the maximum elongation (strain ratio) rate was in the order of SPRG > AR > UR > ER. Comparison of the elongation rate warranted a further investigation into the SPRG material that has a viscoelastic property.(2)A secondary testing followed consisting of measuring the adhesive strength of 3 different types of SPRG, each with different rubber mixture component ratio. For each, a stress-strain diagram was obtained by the inflection point, and an overall stress-strain curve is used to propose a tentative representative yield stress curve for SPRG materials.(3)Lastly, a subsequent experiment by repeated tensile stress application was conducted to derive a sequential stress-strain curves, whereby toughness calculation allowed an assessment of whether the SPRG materials are able to maintain adequate durability against common levels of water pressure compliant to standards in Korea. An example application of durability assessment is proposed by an Abaqus based Finite element method analysis, where the modelling results show that the material is able to withstand a common level of hydrostatic pressure of up to 0.2 N/mm^2^. Varying ranges of hydrostatic pressure conditioning and material property changes can be applied to assess different types of SPRGs currently in the market.

Based on these results, the study proposes the use of a quantitative evaluation method to evaluate the material quality of SPRGs currently used in the market.

## Figures and Tables

**Figure 1 materials-14-07599-f001:**
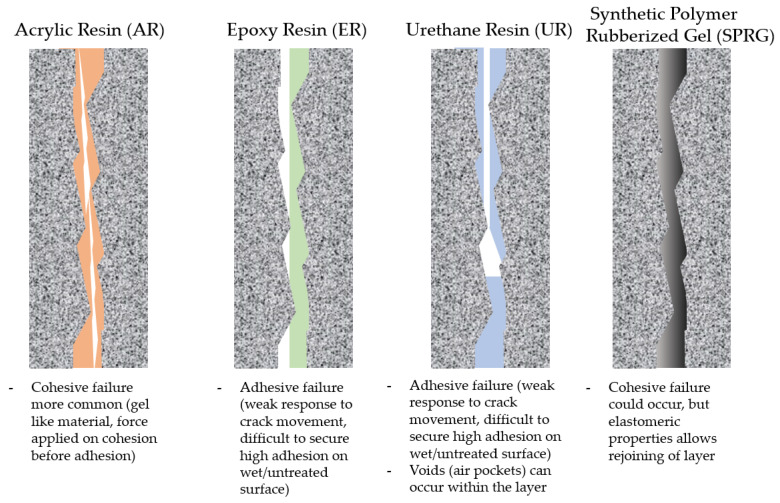
Failure types and mechanism of GILRM illustrated.

**Figure 2 materials-14-07599-f002:**
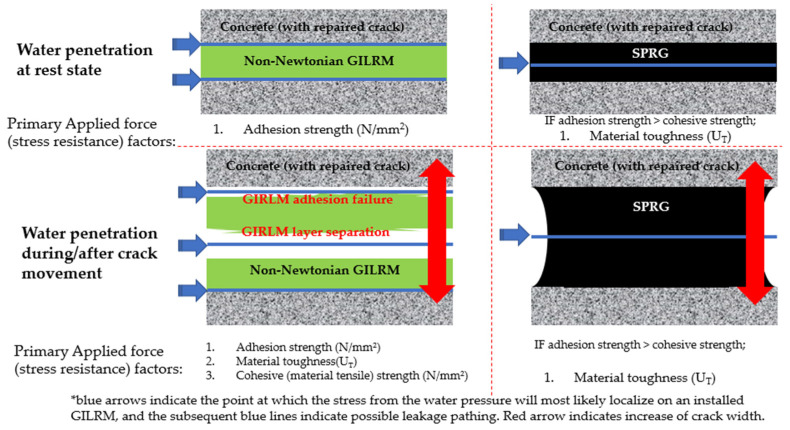
Failure types and mechanism of GILRM illustrated.

**Figure 3 materials-14-07599-f003:**
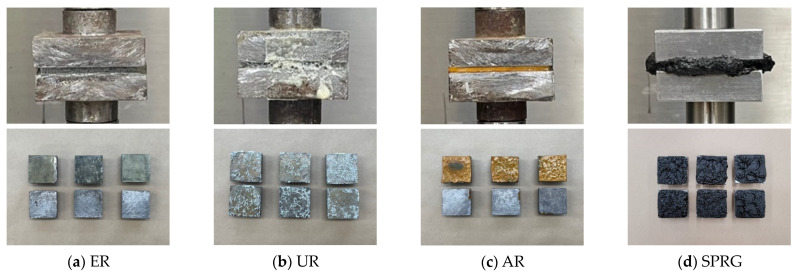
Adhesion testing of each GILRM specimen type.

**Figure 4 materials-14-07599-f004:**
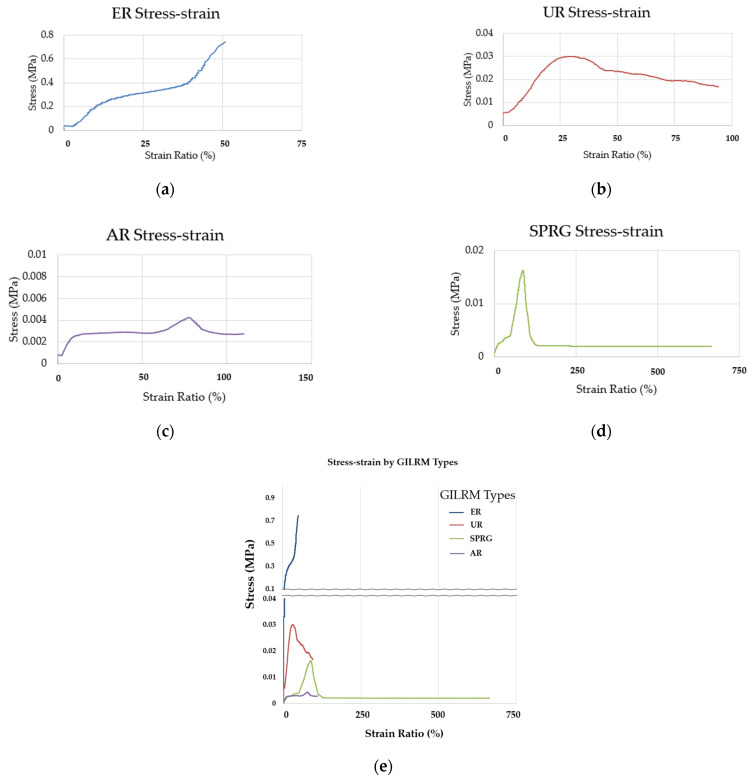
Adhesive stress-strain curve for each material, (**a**) ER results, (**b**) UR results, (**c**) AR results, (**d**) SPRG results, (**e**) comprehensive results (all GILRM types).

**Figure 5 materials-14-07599-f005:**
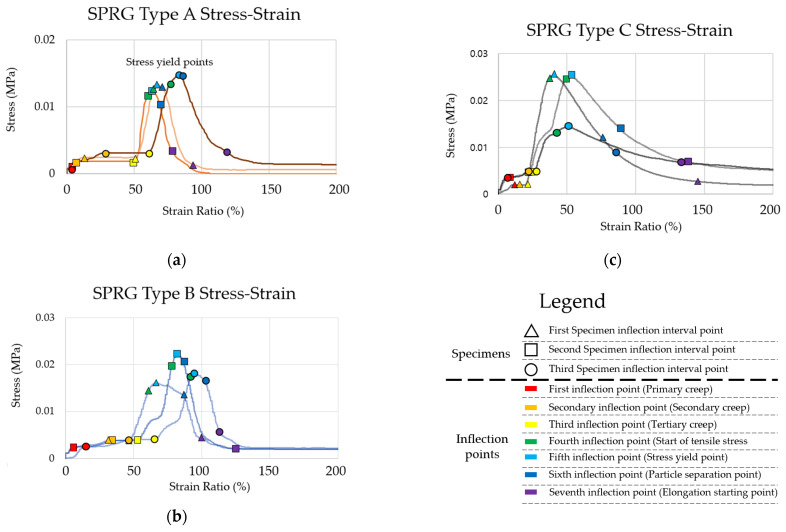
Stress-strain curve according to test results (**a**) SPRG Type A results, (**b**) SPRG Type B results, (**c**) SPRG Type C results.

**Figure 6 materials-14-07599-f006:**
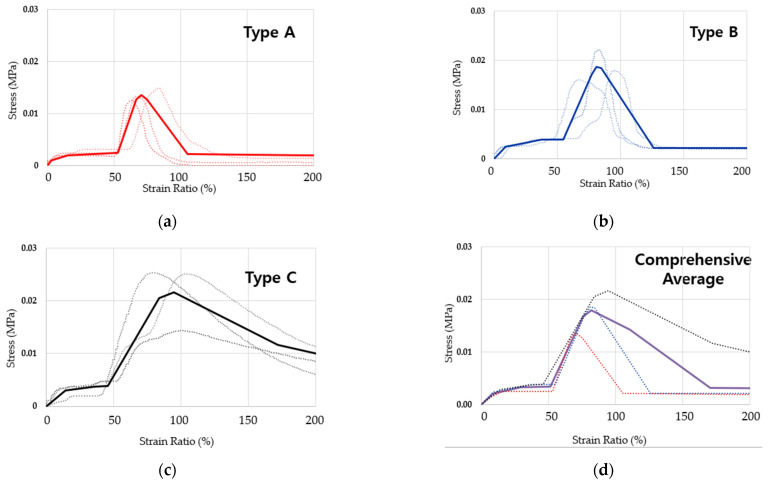
Yield stress derivation based on the 3 types of SPRG average results; (**a**) SPRG Type A averaged result, (**b**) SPRG Type B averaged result, (**c**) SPRG Type C averaged result, (**d**) comprehensive averaged result (Type A, B and C).

**Figure 7 materials-14-07599-f007:**
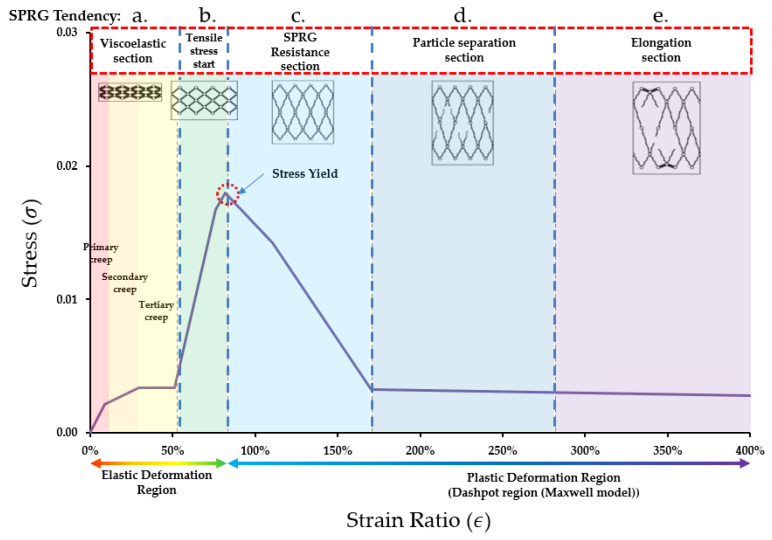
SPRG Stress-Strain curve proposal (tentative).

**Figure 8 materials-14-07599-f008:**
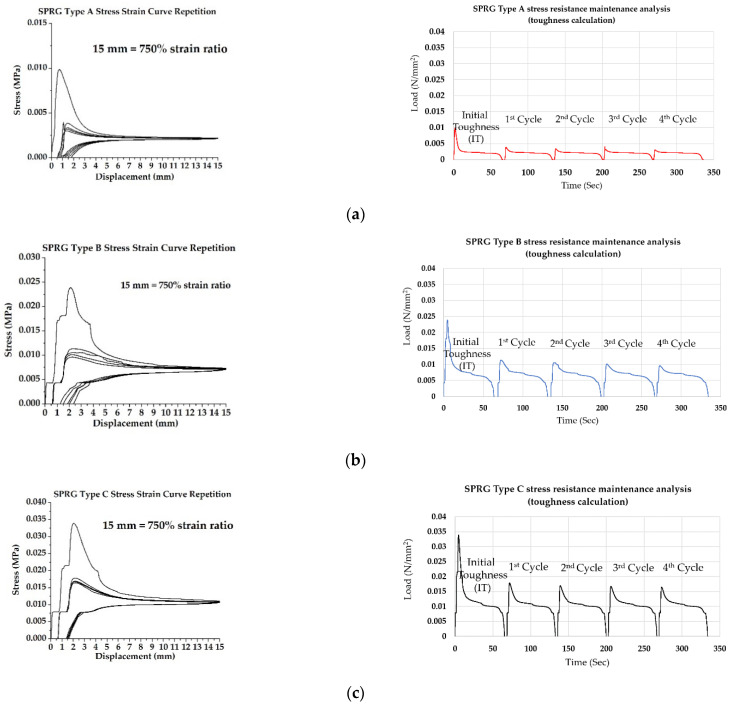
Stress-strain (via load vs. displacement) results from repeated tensile stress application (sample) (**a**) SPRG Type A Stress strain curve repetition and toughness calculation process, (**b**) SPRG Type B Stress strain curve repetition and toughness calculation process, (**c**) SPRG Type C Stress strain curve repetition and toughness calculation process.

**Figure 9 materials-14-07599-f009:**
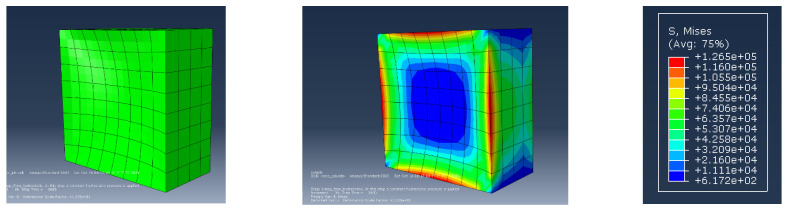
FEM analysis of SPRG based on applicable hydrostatic pressure.

**Table 1 materials-14-07599-t001:** Specimen (material) specification.

Material Type	Composition	Basic Properties
Epoxy Resin (ER)	Epoxy Resin (2): Polyamine (1)	(1) Viscosity: 160 cp/25 °C (2) Curing Reaction: Main component + reagent chemical reaction
Urethane Resin (UR)	Polyurethane resin and other additives	(1) Viscosity: 200–500 cp/25 °C (2) Curing Reaction: reactive to water (3) Volume expansion ratio: >400%
Acrylic Resin (AR)	Sodium polyacrylate, acrylamide-sodium acrylate, Water, hardener (sulfate compound, water), catalyst (triethanolamine, water)	(1) Viscosity: 15 cp/25 °C (2) Curing Reaction: initial reaction with water (3) Solubility: can dissolve in water
Synthetic-polymerized Rubber Gel (SPRG)	Waste rubber, waste oil, tackifier, asphalt, asphalt modifier, filler, etc.	(1) Viscosity: about 3,500,000 cp/25 °C (2) Curing reaction: non-curing (no chemical bonding reaction with water or curing agent) (3) Solidity volume: 95~99%

**Table 2 materials-14-07599-t002:** Composition and viscosity of SPRG.

Properties		Type A	Type B	Type C
Composition	Waste oil	24%	22.5%	20%
	Waste rubber	6%	7.5%	10%
	Asphalt	35%		
	Tackifier	10%		
	Asphalt modifier	5%		
	Filler	20%		
	Total	100%		
Viscosity		1,800,000 cp	3,500,000 cp	5,000,000 cp

**Table 3 materials-14-07599-t003:** Adhesive performance test result of 4 materials.

GILRM Materials	EP	UR	AR	SPRG
Adhesive strength (MPa)	0.6537	0.0381	0.0048	0.0162
Strain at failure (adhesive or cohesive) (%)	51	94	97	664

**Table 4 materials-14-07599-t004:** Adhesion and Elongation measurement results.

Type	Specimen #	Adhesive Strength (N/mm^2^)		Elongation Strain (%)	
Type A	1	0.0125	0.0136	106	695
	2	0.0134		849	
	3	0.0148		1129	
Type B	1	0.0220	0.0186	804	811
	2	0.0179		959	
	3	0.0160		671	
Type C	1	0.0251	0.0216	1881	1667
	2	0.0143		1817	
	3	0.0253		1304	
Avg.		0.0164		1098	

**Table 5 materials-14-07599-t005:** Yield point derivation—Stress (N/mm^2^), Strain (%).

Yield Stress Point Sections		Type A			Type B			Type C		
		1 (Triangle)	2 (Circle)	3 (Square)	1 (Triangle)	2 (Circle)	3 (Square)	1 (Triangle)	2 (Circle)	3 (Square)
		Stress (MPa)								
Inflection Points	Start	0	0	0	0	0	0	0	0	0
	1	0.0012	0.0009	0.0008	0.0023	0.0025	0.0022	0.0034	0.0034	0.0019
	2	0.0018	0.0008	0.0031	0.0038	0.0038	0.0038	0.0045	0.0047	0.0020
	3	0.0018	0.0024	0.0031	0.0038	0.0040	0.0038	0.0046	0.0047	0.0020
	4	0.0117	0.0128	0.0134	0.0195	0.0171	0.0142	0.0242	0.0129	0.0244
	5	0.0125	0.0134	0.0148	0.0220	0.0179	0.0160	0.0251	0.0143	0.0253
	6	0.0104	0.0130	0.0146	0.0216	0.0175	0.0158	0.0245	0.0127	0.0251
	7	0.0035	0.0014	0.0015	0.0021	0.0023	0.0020	0.0054	0.0062	0.0019
	After	0.0001	0.0005	0.0012	0.0019	0.0021	0.0019	0.0025	0.0027	0.0000
		**Strain Ratio (%)**								
Inflection Points	Start	0	0	0	0	0	0	0	0	0
	1	3	3	3	6	15	6	13	10	20
	2	6	12	28	35	47	32	40	41	28
	3	49	50	61	53	66	47	46	53	40
	4	60	64	77	78	92	61	97	83	73
	5	63	66	83	82	95	67	105	100	79
	6	69	70	86	85	99	70	113	128	85
	7	78	93	146	125	131	123	346	327	379
	After	106	849	1129	156	959	671	1881	1817	1304

**Table 6 materials-14-07599-t006:** Average stress yield derived results.

Yield Stress Point Sections	Type A		Type B		Type C		Types A,B,C Avg.	
	Stress (N/mm^2^)	Strain (%)	Stress (N/mm^2^)	Strain (%)	Stress (N/mm^2^)	Strain (%)	Stress (N/mm^2^)	Strain (%)
Start	0.0000	0	0.0000	0	0.0000	0	0.0000	0
1	0.0010	3	0.0023	9	0.0029	14	0.0026	15
2	0.0019	15	0.0038	38	0.0037	36	0.0033	34
3	0.0024	53	0.0039	55	0.0038	46	0.0034	46
4	0.0126	67	0.0169	77	0.0205	84	0.0186	78
5	0.0136	70	0.0186	81	0.0216	95	0.0198	90
6	0.0127	75	0.0183	85	0.0207	108	0.0189	106
7	0.0021	105	0.0021	126	0.0045	351	0.0040	353
Yield stress Point	0.0006	695	0.0020	811	0.0017	1667	0.0014	1560

**Table 7 materials-14-07599-t007:** Toughness calculation—Load (N/mm^2^), Displacement (mm).

Conditions				Toughness Per Specimen (N/mm^2^)									
SPRG Types		Type A				Type B				Type C			
Specimen Number		1	2	3	Average	1	2	3	Average	1	2	3	Average
StressPer Cycle	InitialToughness	88.7	75.7	63.5	76.0	248.1	207.5	292.1	249.2	1462.6	1707.5	1220.6	1463.6
	1st Cycle	63.5	54.2	45.5	54.4	186.5	156.0	219.7	187.4	275.6	321.8	230.0	275.8
	2nd Cycle	61.9	52.8	44.4	53.0	183.4	153.4	216.0	184.3	269.5	314.6	224.9	269.7
	3rd Cycle	61.3	53.1	44.2	52.9	177.5	148.5	209.0	178.3	271.5	317.0	226.6	271.7
	4th Cycle	60.9	52.0	43.7	52.2	171.9	143.8	202.4	172.7	268.1	312.9	223.7	268.2
	Average	61.9	53.0	44.5	53.1	179.8	150.4	211.8	180.7	271.2	316.6	226.3	271.4

## Data Availability

Not applicable.

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
