# Peer review of "Crack-Bridging Property Evaluation of Synthetic Polymerized Rubber Gel (SPRG) through Yield Stress Parameter Identification"

_materials, 2021, doi:10.3390/ma14247599_

Round 1

Reviewer 1 Report

This paper is not well written. The structure is not clear. Therefore, I believe it should not be considered for publication in this journal.

Following are some comments that might be useful for further improvement.

  • Both the abstract and the introduction are made up of long and not very objective sentences. The writing of the manuscript does not resemble that of a technical text. Furthermore, it is repetitive regarding the superior characteristics of SPRG compared to conventional GILRMs. The text lacks English revision and objectivity.
  • The sentence "As shown in Figure 1, when comparing the viscosities of general liquid materials, the viscosity range of construction polymer materials used for structural leak repair among materials for food, industry, and construction materials is very high, and among them, SPRG can be seen to have the highest viscosity within the category" is a clear example of the perceived lack of objectivity in the text. The sentence is too long and difficult to assimilate the content. In addition, it uses the word "materials" four times.
  • The text does not present a coherent division: Introduction, objectives, methodology (materials and methods), results and conclusions.
  • I believe that in the methodology it is interesting to first present the materials to be tested (and their properties). Only afterwards, the proposed test method for measuring the stress-strain of the GILRMs. The font size in Figure 5 needs to be changed.
  • Figure 6 shows the failure mode (result) of the different materials tested. Therefore, it should not be presented in the work methodology.
  • The text does not present a coherent division: Introduction, objectives, methodology (materials and methods), results and conclusions.
  • The FEM analysis carried out in the work are presented right before the conclusions. These analyzes are not cited in the summary and introduction of the document.
  • The conclusions need to be refined. Authors should avoid overuse of results in this section. Furthermore, it must strictly follow the order of the objectives proposed in the study.

Author Response

The authors of article materials–1450355 would like to extend their sincerest gratitude to the reviewers for taking time out of their busy schedule to review our paper. Much thanks to your efforts, comments and contribution, we hope that the paper has improved in quality and clarity. The below lists the comments provided by the reviewers and the respective response and details as to how the comments were address and applied in the revised article.

 Reviewer comment 1 

Both the abstract and the introduction are made up of long and not very objective sentences. The writing of the manuscript does not resemble that of a technical text. Furthermore, it is repetitive regarding the superior characteristics of SPRG compared to conventional GILRMs. The text lacks English revision and objectivity.

Response 1

  1. The abstract and the introduction were revised to be more objective
  2. The semantics of the manuscript has been revised to provide a more objective and technical text. Parts of the manuscript that seem to unnecessarily promote the properties of SPRG were revised to maintain a more objective stance.

Please refer to the revised manuscript lines 22-39 (abstract) and lines 46-93 (introduction) for details.

Reviewer comment 2

The sentence "As shown in Figure 1, when comparing the viscosities of general liquid materials, the viscosity range of construction polymer materials used for structural leak repair among materials for food, industry, and construction materials is very high, and among them, SPRG can be seen to have the highest viscosity within the category" is a clear example of the perceived lack of objectivity in the text. The sentence is too long and difficult to assimilate the content. In addition, it uses the word "materials" four times.

Response 2

The original section 2.1 has been removed from the manuscript. Upon review, the section was largely irrelevant and did contribute towards the goals of the manuscript content.

Reviewer comment 3

The text does not present a coherent division: Introduction, objectives, methodology (materials and methods), results and conclusions.

I believe that in the methodology it is interesting to first present the materials to be tested (and their properties). Only afterwards, the proposed test method for measuring the stress-strain of the GILRMs.

Response 3

The article sections have been renamed accordingly as suggested by the reviewer. The contents of the sections have been rearranged and revised accordingly such that the division of the sections are in order. Sections that discuss material properties and composition have been placed in Section 3.1 (Lines 194-220) in the revised manuscript.

Reviewer comment 4

The font size in Figure 5 needs to be changed.

 Response 4

Font size of Figure 5 has been changed such that the texts are now clearer.

Reviewer comment 5

Figure 6 shows the failure mode (result) of the different materials tested. Therefore, it should not be presented in the work methodology.

 Response 5

Figure 6 (now Figure 5 in the revised manuscript) has been moved to Section 3.1 ‘proposed test method for measuring the stress-strain of the tested GILRMs,’ to be shown after Figure 5.

Reviewer comment 6

The FEM analysis carried out in the work are presented right before the conclusions. These analyzes are not cited in the summary and introduction of the document.

Response 6

FEM analysis results and significance has been included in the abstract and the introduction of the article. The FEM analysis is not the core conclusion and is presented to demonstrate an application and usefulness of the evaluation methodology proposed in this study, and is just provided as a suggestion rather than a connected result of the experiment. A detailed analytical results have been provided in Lines 441-446, and the correlation to FEM analysis to the experimental study have been outlined in the conclusion section.

Reviewer comment 7

The conclusions need to be refined. Authors should avoid overuse of results in this section. Furthermore, it must strictly follow the order of the objectives proposed in the study.

Response 7

The conclusion section has been revised substantially to be more clear about the findings of this study;

This study was conducted to propose a new evaluation method for GILRM based on stress-strain analysis using standard adhesion strength testing method. The following conclusions were drawn;

(1) A preliminary testing consisting of adhesive strength measurement testing of four types of GILRM (UR, ER, AR and SPRG) was conducted. Adhesive strength of the four GILRM appears in the order of AR<SPRG<UR<ER. A Stress-strain diagram for each GILRM specimen types based on the measurement results, whereby it was confirmed that the maximum elongation (strain ratio) rate was in the order of SPRG>AR>UR>ER. Comparison of the elongation rate warranted a further investigation into the SPRG material that has a viscoelastic property.

(2) A secondary testing followed consisting of measuring the adhesive strength of 3 different types of SPRG, each with different rubber mixture component ratio. For each, a stress-strain diagram was obtained by the inflection point, and an overall stress-strain curve is used to propose a tentative representative stress yield curve for SPRG materials.

(3) Lastly, a subsequent experiment by repeated tensile stress application was conducted to derive a sequential stress-strain curves, whereby toughness calculation allowed an assessment of whether the SPRG materials are able to maintain adequate durability against common levels of water pressure compliant to standards in Korea. An example application of durability assessment is proposed by an Abaqus based Finite element method analysis, where the modelling results show that the material is able to withstand a common level of hydrostatic pressure of up to 0.2 N/mm2. Varying ranges of hydrostatic pressure conditioning and material property changes can be applied to assess different types of SPRGs currently in the market.

Based on these results, the study proposes the use of a quantitative evaluation method to evaluate the material quality of SPRGs currently used in the market.

Please refer to the revised manuscript Lines 455 to 475 for details.

The authors would like to express their thanks once again for the reviewer’s time and valuable feed back on this manuscript.

Reviewer 2 Report

The authors should improve the introduction and the bibliography citing other recent and relevant research papers presented in international journals dealing with this topic.

Please see the attached comments.

Author Response

The authors of article materials–1450355 would like to extend their sincerest gratitude to the reviewers for taking time out of their busy schedule to review our paper. Much thanks to your efforts, comments and contribution, we hope that the paper has improved in quality and clarity. The below lists the comments provided by the reviewers and the respective response and details as to how the comments were address and applied in the revised article.

Reviewer comment 1 

Are the results interpreted appropriately?

This study is characterized by an high level of originality and scientific soundness. It is presented in a really appropriate way, both technically and scientifically. The conclusions are supported the quantity and quality of the data presented and by the results.

Response 1

The authors sincerely appreciate the reviewer’s comment.

Reviewer comment 2 

The figures should be improved: some images are not clearly visible, but in general the methods and the tools are described in a way that other researchers can reproduce the results.

Response 2

The figures have been revised throughout the article such that the texts (Figure 4, 9 and 10) and images are more clear for the readers.

Reviewer Comment 3

The authors should improve the introduction and the bibliography citing other recent and relevant research papers presented in international journals dealing with this topic.

Response 3

The abstract and the introduction were revised to be more objective

The semantics of the manuscript has been revised to provide a more objective and technical text. Parts of the manuscript that seem to unnecessarily promote the properties of SPRG were revised to maintain a more objective stance.

Please refer to the revised manuscript lines 22-39 (abstract) and lines 46-93 (introduction) for details.

The bibliography list has been extended as well. However, it must be noted that there are not many relevant research papers regarding this topic. Studies on individual and specific grout injection material types and their characteristics can be found, but topics on renewal/development/amendment of existing test method by proposing a new evaluation criteria does not exist, as viscoelastic type of GILRM are relatively new type of material.

Reviewer comment 4

In Table 6 the unit of measurements are not clearly visible, please improve;

Response 4

Table 6 units of measurement has been revised

Reviewer comment 5

 Please check the font of the tables. Smaller fonts may be used, but no less than 8 pt. in size;

Response 5

The font sizes of the tables have been checked and revised such that none of the fonts throughout the article are no less than 8 pt. in size.

Reviewer comment 6

To make a clearer reading of the paper please put the words “Figure” and “Table” in text and in the captions in bold style;

Response 6

Reference wordings in the text for Figures and Tables have been captioned in bold style.

Reviewer comment 7

 - Please check the references, in particular for the journals. They should be described as follows: 1. Author 1, A.B.; Author 2, C.D. Title of the article. Abbreviated Journal Name Year, Volume, page range.

Response 7

Reference formatting has been revised to follow the regulations of the MDPI template.

The authors would like to express their thanks once again for the reviewer’s time and valuable feed back on this manuscript.

Reviewer 3 Report

This study investigates the stress yield derivation by stress-strain curve analysis on four various grout injection leakage repair materials such as Acrylic, epoxy, urethane and SPRG. The study is pretty helpful for repairing long-term safety maintenance of concrete structures. However, it should be improved further by addressing the following comments.

 1. Abstract is too extended and an explanation of the study conducted is not in detail. Try to include more details about the study conducted.

2. Present only key findings in the abstract. The current findings are more extended and hard to follow.

3. Include the citations for the ISO TR 16475, ISO TS 16774 and other ISO’s, if any.

4. Figure 3, are the failure type “cohesive failure” for the Acrylic resin and SPRG are identical and ER and UR show a different failure. Highlight the reason for these failures in section 2.2.

5. Page 10; waste rubber content can change the adhesive strength. What is the limitation of waste rubber content? Since it has mentioned three rubber content in the article (6, 7.5 and 10%).

6. Table 6, Type A, B and C contain three content of rubber which indicated different performances. What signifies the average of three results since the rubber content is different?

7. Page 12, the discussions of each paragraph are the same with only changing the values. It would be better to paraphrase to avoid repetition.

8. Figure 11 can be improved. It is hard to read the texts in the figure.

9. Page 15, Include the Equation to calculate the initial toughness.

10. More details about the finite element model should be included in detail.

11. Compare the results of the experimental and finite element methods.

12. More recent literatures (2020, 2021) can be included to strengthen the article further.

Author Response

The authors of article materials–1450355 would like to extend their sincerest gratitude to the reviewers for taking time out of their busy schedule to review our paper. Much thanks to your efforts, comments and contribution, we hope that the paper has improved in quality and clarity. The below lists the comments provided by the reviewers and the respective response and details as to how the comments were address and applied in the revised article.

This study investigates the stress yield derivation by stress-strain curve analysis on four various grout injection leakage repair materials such as Acrylic, epoxy, urethane and SPRG. The study is pretty helpful for repairing long-term safety maintenance of concrete structures. However, it should be improved further by addressing the following comments.

Reviewer comment 1

Abstract is too extended and an explanation of the study conducted is not in detail. Try to include more details about the study conducted.  Present only key findings in the abstract. The current findings are more extended and hard to follow.

Response 1

The abstract and the introduction were revised to be more objective

The semantics of the manuscript has been revised to provide a more objective and technical text. Parts of the manuscript that seem to unnecessarily promote the properties of SPRG were revised to maintain a more objective stance.

Please refer to the revised manuscript lines 22-39 (abstract) and lines 46-93 (introduction) for details.

The bibliography list has been extended as well. However, it must be noted that there are not many relevant research papers regarding this topic. Studies on individual and specific grout injection material types and their characteristics can be found, but topics on renewal/development/amendment of existing test method by proposing a new evaluation criteria does not exist, as viscoelastic type of GILRM are relatively new type of material.

Reviewer comment 2

Include the citations for the ISO TR 16475, ISO TS 16774 and other ISO’s, if any.

Response 2

Citation for the ISO documents have been included in the bibliography in the revised manuscript (please refer to bibliography list number 4 and 5 in the revised version of the manuscript)

Reviewer comment 3

Figure 3, are the failure type “cohesive failure” for the Acrylic resin and SPRG are identical and ER and UR show a different failure. Highlight the reason for these failures in section 2.2.

Response 3

Figure 3 (Now Figure 2 in the revised manuscript) has been revised to be more clear about the respective failure types and their explanations. More detailed explanation was also added at the end of section 2.2 (Please refer to the revised manuscript, Lines 135-141)

Reviewer comment 4

Page 10; waste rubber content can change the adhesive strength. What is the limitation of waste rubber content? Since it has mentioned three rubber content in the article (6, 7.5 and 10%). Table 6, Type A, B and C contain three content of rubber which indicated different performances. What signifies the average of three results since the rubber content is different?

Response 4

Explanation and reasoning on the comparison and contrast of three different “types” of SPRG with different rubber content have been added in the manuscript; to paraphrase, at the moment there is no standard or regulation for a consistent mixture ratio for SPRG manufacturing companies. The fundamental properties are the same for all these “types” in that they are all modified asphalt based rubberized compounds with different fillers. The problem is that some companies, in order to secure new patents or receive funding approval from governments, make low performance but eco-friendly version of the SPRG that have lower potential than other existing product versions, while some are purely performance based without consideration of workability (viscosity is far too high for workers to use). As all of these products cannot be neglected, and at this point we as authors are tasked to provide at least a starting point for characterizing this material, we thought to average out one of the highest viscosity type and lowest viscosity type and attempt to provide an overall yield stress curve graph, as we need a point of a reference to start the evaluation regime for the repeated crack movement resistance performance. Using only the lowest viscosity SPRG or the highest would lack the proper representation of this material. The authors revised the text to specifically mention that this yield-stress curve proposal is tentative and is subject to change in the future once a more standardized composition of SPRG is set.

SPRG used in the experiment for this analysis was a synthetic rubber-based material obtained by thermally fusion of waste oil and waste rubber. SPRGs high viscoelastic material manufactured into liquid rubber by finely pulverizing the collected and processed waste rubber for recycling to make powder with a particle size of 200 to 400 μm, and then thermally fused with waste oil. In the case of Korea, a standardized composition ratio of SPRG has not been developed [27] and this leads to case of SPRG products with different rubber mixture ratios being used in the market. While this does not fundamentally change the characteristic of SPRG, higher viscosity has traditionally proven to provide higher adhesion strength but more difficult workability and vice versa. As there is a large range of these compositions, for the experimental group in this study, the contents of waste oil and waste rubber, which are the main materials that give the adhesiveness of synthetic rubber materials, were 4:1 (low viscosity, 1.8 million cp), 3:1 (medium viscosity, 3.5 million cp), and 2:1 (high viscosity), 5 million cp) in accordance to 3 most commonly used SPRG products currently being used in Korea. (Lines 205 to 217)

A stress-strain curve was calculated based on the measurement results of adhesive strength and elongation ratio of Types A to C. The reason for a comprehensive comparison of SPRG types of different viscosities is due to the current factor related to a lack of standardization of SPRG mixture composition. As has been explained in section 3.1, higher rubber composition ratio that results in higher adhesive strength has not always resulted in positive results in terms of overall installation quality of SPRG materials. As a means to incorporate the variation of different types and viscosity ranges of SPRG materials, this study will tentatively propose a yield stress curve analysis based the average of the 3 different types of SPRG. It must be noted that further investigation by a more standardized composition of SPRG will provide different yield stress curve limit in future studies. (Lines 298 to 306)

Reviewer comment 5

 Page 12, the discussions of each paragraph are the same with only changing the values. It would be better to paraphrase to avoid repetition.

Response 5

Discussions of the results have been paraphrased (please refer to lines 259 to 277 in the revised version of the manuscript)

Reviewer comment 6

Figure 11 can be improved. It is hard to read the texts in the figure.

Response 6

Figure 11 has been revised (now Figure 9 in the revised version of the manuscript)

Reviewer comment 7

Page 15, Include the Equation to calculate the initial toughness.

Response 7

Equation (2) for calculating the initial toughness (and is the same for the cycles) has been included in section 4.3, Lines 404-414

Reviewer comment 8

More details about the finite element model should be included in detail. Compare the results of the experimental and finite element methods.

Response 8

Details about the FEM model has been included in the manuscript, but it must be noted that the FEM analysis is not the core conclusion and is presented to demonstrate an application and usefulness of the evaluation methodology proposed in this study, and is just provided as a suggestion rather than a connected result of the experiment. A detailed analytical results have been provided in Lines 441-446, and the correlation to FEM analysis to the experimental study have been outlined in the conclusion section;

By referencing this principle, the energy exerted by water pressure (calculable by Equation 1 in Section 2.2) based on the depth of the concrete structure can be calculated and be employed as a criteria index for the tested SPRG; as long as the toughness (energy) of the SPRG is higher than that of the expected water pressure force, the said material can be used, and higher toughness index can be used as an index of waterproofing performance of the SPRG. This is applicable for finite element method (FEM) analysis (via Abaqus) by the method of static hydrostatic pressure simulation of a viscoelastic model of an SPRG sample. While further analysis is still required, based on the FEM analysis, the types of SPRG can be subject to a fixed hydrostatic pressure compliant to the pressure levels common to underground structure environment (in the case of Korea, 20 m underground equates to 0.2 N/mm2) (Refer to Figure 10 below for illustration of the modelling samples), which when applied upon the model based on the dimensions of the specimen size (40 × 40 × 2 mm), the derived strain on the material can be used to calculate the corresponding stress based on the results of Figure 10. In this sample case, based on the results of a sample modelling of Type A SPRG, the maximum deformation derived at the edges (shown as red sections in the image) is converted to a unit of 1.265 mm, where in comparison to the yield stress range, the material would not have undergone cohesive failure due to the given hydrostatic pressure.

Reviewer comment 9

 More recent literatures (2020, 2021) can be included to strengthen the article further.

Response 9

 The bibliography list has been extended as well. However, it must be noted that there are not many relevant research papers regarding this topic. Studies on individual and specific grout injection material types and their characteristics can be found, but topics on renewal/development/amendment of existing test method by proposing a new evaluation criteria does not exist, as viscoelastic type of GILRM are relatively new type of material. Please refer to the revised bibliography section of the new manuscript

The authors would like to express their thanks once again for the reviewer’s time and valuable feed back on this manuscript.

Reviewer 4 Report

1. The title should be abbreviated and abbreviated.
2. In the abstract of the technical article, add the main numerical results.
3. There is no aim and tasks to achieve it. Accordingly, in the conclusions it is necessary to give a numbered list on the achievement of the assigned tasks.
4. The literature review should be expanded with modern articles on building materials. For example:
- Svintsov, A.P., Shchesnyak, E.L., Galishnikova, V.V. Fediuk, R.S., Stashevskaya, N.A. Effect of nano-modified additives on properties of concrete mixtures during winter season. Construction and Building Materials. 2020.237, 117527.https: //doi.org/10.1016/j.conbuildmat.2019.117527
- Semenov, PA; Uzunian, AV; Davidenko, AM et al. First study of radiation hardness of lead tungstate crystals at low temperatures. Nuclear instruments & methods in physics research section A: Accelerators spectrometers detectors and associated equipment. 2007.582 (2). pp. 575-580. DOI: 10.1016 / j.nima.2007.08.178
- Elistratkin, M.Y., Lesovik, V.S., Zagorodnjuk, L. H. New point of view on materials development. IOP Conference Series Materials Science and Engineering 2018.327 (3), 032020 DOI: 10.1088 / 1757-899X / 327/3/032020

Author Response

The authors of article materials–1450355 would like to extend their sincerest gratitude to the reviewers for taking time out of their busy schedule to review our paper. Much thanks to your efforts, comments and contribution, we hope that the paper has improved in quality and clarity. The below lists the comments provided by the reviewers and the respective response and details as to how the comments were address and applied in the revised article.

Reviewer comment 1

The title should be abbreviated and abbreviated.

Response 1

Title has been abbreviated; “Crack-bridging property evaluation of synthetic polymerized rubber gel (SPRG) through stress yield parameter identification”

Reviewer comment 2
In the abstract of the technical article, add the main numerical results.

Response 2

The abstract and the introduction were revised to be more objective

The semantics of the manuscript has been revised to provide a more objective and technical text. Parts of the manuscript that seem to unnecessarily promote the properties of SPRG were revised to maintain a more objective stance.

Please refer to the revised manuscript lines 22-39 (abstract) and lines 46-93 (introduction) for details.

Reviewer comment 3

There is no aim and tasks to achieve it. Accordingly, in the conclusions it is necessary to give a numbered list on the achievement of the assigned tasks.

Response 3

The conclusion section has been revised substantially to be more clear about the findings of this study;

This study was conducted to propose a new evaluation method for GILRM based on stress-strain analysis using standard adhesion strength testing method. The following conclusions were drawn;

(1) A preliminary testing consisting of adhesive strength measurement testing of four types of GILRM (UR, ER, AR and SPRG) was conducted. Adhesive strength of the four GILRM appears in the order of AR<SPRG<UR<ER. A Stress-strain diagram for each GILRM specimen types based on the measurement results, whereby it was confirmed that the maximum elongation (strain ratio) rate was in the order of SPRG>AR>UR>ER. Comparison of the elongation rate warranted a further investigation into the SPRG material that has a viscoelastic property.

(2) A secondary testing followed consisting of measuring the adhesive strength of 3 different types of SPRG, each with different rubber mixture component ratio. For each, a stress-strain diagram was obtained by the inflection point, and an overall stress-strain curve is used to propose a tentative representative stress yield curve for SPRG materials.

(3) Lastly, a subsequent experiment by repeated tensile stress application was conducted to derive a sequential stress-strain curves, whereby toughness calculation allowed an assessment of whether the SPRG materials are able to maintain adequate durability against common levels of water pressure compliant to standards in Korea. An example application of durability assessment is proposed by an Abaqus based Finite element method analysis, where the modelling results show that the material is able to withstand a common level of hydrostatic pressure of up to 0.2 N/mm2. Varying ranges of hydrostatic pressure conditioning and material property changes can be applied to assess different types of SPRGs currently in the market.

Based on these results, the study proposes the use of a quantitative evaluation method to evaluate the material quality of SPRGs currently used in the market.

Please refer to the revised manuscript Lines 455 to 475 for details.

Reviewer comment 4

The literature review should be expanded with modern articles on building materials.

Response 4

 The bibliography list has been extended as well with the inclusion of the proposed articles. However, it must be noted that there are not many relevant research papers regarding this topic. Studies on individual and specific grout injection material types and their characteristics can be found, but topics on renewal/development/amendment of existing test method by proposing a new evaluation criteria does not exist, as viscoelastic type of GILRM are relatively new type of material. Please refer to the revised bibliography section of the new manuscript

The authors would like to express their thanks once again for the reviewer’s time and valuable feed back on this manuscript.